# BulletGen: Improving 4D Reconstruction with Bullet-Time Generation

## Abstract

Transforming casually captured, monocular videos into fully immersive dynamic experiences is a highly ill-posed task, and comes with significant challenges, *e.g.*, reconstructing unseen regions, and dealing with the ambiguity in monocular depth estimation. In this work, we introduce BulletGen, an approach that takes advantage of generative models to correct errors and complete missing information in a Gaussian-based dynamic scene representation. This is done by aligning the output of a diffusion-based video generation model with the 4D reconstruction at a single frozen "bullet-time" stamp. The generated frames are then used to supervise the optimization of the 4D Gaussian model. Our method seamlessly blends generative content with both static and dynamic scene components, achieving state-of-the-art results on both novel-view synthesis, and 2D/3D tracking tasks.

## 1 Introduction

The task of 3D reconstruction from color images is fundamental to computer vision, with a variety of solutions being proposed over the years (Kerbl et al., 2023; Mildenhall et al., 2020; Schönberger & Frahm, 2016; Schönberger et al., 2016). While impressive strides have been made towards improving both the completeness and accuracy of results, most methods and datasets focus exclusively on static scenes. However, the real world is dynamic, and the ability to reconstruct dynamic content opens up many new opportunities in immersive media generation and robotics.

The reconstruction of dynamic scenes presents a very challenging 4D problem, as it requires reasoning about both geometry and motion of the scene. This is exacerbated by the fact that traditional 4D reconstruction methods require capturing multi-view images of dynamic scenes with expensive camera rigs (Orts-Escolano et al., 2016; Joo et al., 2015), resulting in limited amount of useful datasets for algorithm development and evaluation. Therefore, building on developments in novel 3D scene representations (Kerbl et al., 2023; Mildenhall et al., 2020), the attention of the research community in recent years has focused on 4D reconstruction from single-view monocular videos (Luiten et al., 2024; Park et al., 2021a; Wu et al., 2024a). But, as a monocular video only observes the scene from a single viewpoint at any given timestep, the problem of 4D reconstruction in this context is heavily under-constrained. As a consequence, existing methods can only find locally optimal solutions, and fail when synthesizing novel views substantially deviating from the training set views (Fig. 1).

Motivated by breakthroughs in learning priors from large-scale data, many recent methods have sought to mitigate the local minima problem by leveraging generative models – particularly diffusion-based methods (Ren et al., 2025; Sun et al., 2024; Van Hoorick et al., 2024) – to constrain the output to lie in the distribution of natural images. However, in the majority of cases, these methods only predict 2D projections of a dynamic scene, and therefore fail to take advantage of the efficient rendering and global consistency advantages provided by 3D scene representations that are optimized per-scene, such as Neural Radiance Fields (NeRFs) (Mildenhall et al., 2020) and 3D Gaussian Splatting (3DGS) (Kerbl et al., 2023). In this paper, we propose a method to incorporate these 2D projections into a consistent and plausible 4D reconstruction.

Our method, termed *BulletGen*, uses a video diffusion model to generate novel views for selected frozen time stamps – so-called bullet times (Liang et al., 2024; Wang et al., 2021). The diffusion model is conditioned on a rendered frame along with natural language image captions from the input video. The generated views are then used to iteratively supervise a global 3D representation. This latter step is achieved by accurately tracking and aligning the generated views to the input coordinate frame, and

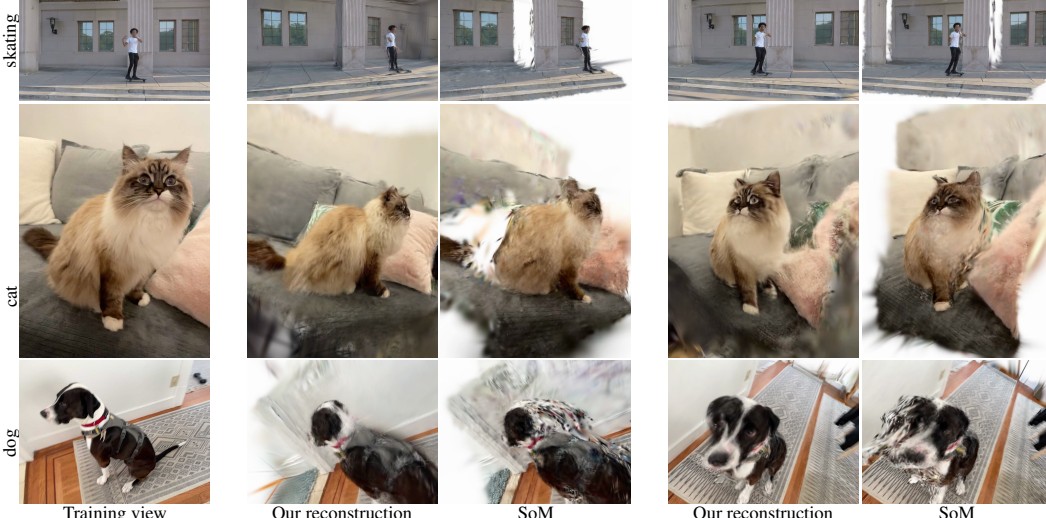

Figure 1: **Extreme novel view synthesis** of a 4D scene with generative model guidance in frozen-time instances (bullet times). The input is only a monocular video on the left. We compare to the Shape-of-Motion (SoM) (Wang et al., 2024b) method on the *cat* and *dog* sequences from the iPhone dataset (Gao et al., 2022) and the *skating* sequence from Nvidia dataset (Yoon et al., 2020).

by using a novel reconstruction loss based on photometric, perceptual, semantic, and depth errors. By repeating this process for different time stamps, the initially inconsistent 2D projections are robustly incorporated into a consistent dynamic 3D reconstruction. This is similar to other vision tasks (multi-view stereo, SLAM, bundle adjustment) that also successfully integrate independent predictions through principled optimization. We use 3D Gaussians as the underlying scene representation, but our method is general and could work with any differentiable 3D representation.

In summary, we make the following contributions:

- We propose a method for dynamic 3D scene reconstruction from monocular RGB videos by augmenting the training views using a generative video diffusion model at selected bullet times. Our bullet-time static diffusion strategy uniquely leverages abundant static training data, making it more practical than methods requiring dynamic video training. Our method provides a scalable solution for monocular 4D reconstruction, avoiding the computational burden of training dynamic diffusion models while achieving comparable or better results.
- We compare our method against prior work, and show that BulletGen achieves state-of-the-art results on several benchmark datasets for novel view synthesis quality, and 2D/3D tracking accuracy.
- The reconstructed dynamic scene also contains new synthesized parts of the scene that seamlessly and plausibly blend in the original scene for both the static (*e.g.* new pillows for *cat* and generated walls for *skating* in Fig. 1) and the dynamic parts (*e.g.* backside of the *cat*, the full head of the *dog*).

## 2 RELATED WORK

The problem of 3D reconstruction and novel-view synthesis has a long history in the fields of computer vision and graphics. Early approaches used dense multi-view images (Davis et al., 2012; Kim et al., 2013; Wilburn et al., 2005) to capture a spatio-angular *light fields* (Gortler et al., 2023; Levoy & Hanrahan, 2023), which could be re-parameterized to generate novel views via quadrilinear interpolation. However, the number of views required by sampling theory to enable such interpolation has been shown to be exorbitantly high (Chai et al., 2000). As a result, novel view synthesis remained a niche task requiring expensive camera rigs, specialized sensors, or time-consuming capture setups (Davis et al., 2012; Georgiev et al., 2006; Ng et al., 2005; Wilburn et al., 2005). Mildenhall et al. (2019) and Zhou et al. (2018) were among the first to seek to overcome the sampling limits using learned data priors encoded as the weights of a convolutional neural network. Subsequent work built on this work by exploiting the ever-increasing capabilities of deep neural networks to not

only lower the sampling requirements for view synthesis, even down to a single image (Chen et al., 2024; Han et al., 2022; Khan et al., 2023; Liu et al., 2023a;b; Long et al., 2024; Tucker & Snavely, 2020; Zhang et al., 2024b), but also proposed new representations that could be optimized per scene to achieve exceptionally high quality and improved spatio-temporal consistency by exploiting scene-specific correlations across views (Luiten et al., 2024; Li et al., 2024a; Kirillov et al., 2023; Liang et al., 2025a). These two research directions naturally converge on the task of 4D reconstruction from monocular videos. However, in this context, past work that has used per-scene optimization to track and correlate points across frames has often failed to take advantage of the strong data priors of pre-trained models specifically for view synthesis.

**Per-scene monocular reconstruction.** Following the seminal work of Mildenhall et al. (2020) on static reconstruction by optimizing a scene-specific 3D representation by Neural Radiance Fields (NeRFs), many works have adopted a similar strategy to exploit more global context for the problem of 4D reconstruction from a monocular video (Cao & Johnson, 2023; Fridovich-Keil et al., 2023; Li et al., 2021; Park et al., 2021a;c; Pumarola et al., 2021; Shao et al., 2023; Stearns et al., 2024; Yang et al., 2023b). The underlying representation often models the scene as an implicit field (Xie et al., 2022) parameterized by time, which is optimized using the input frames as supervision. The use of a single globally consistent representation, which also implicitly encodes a local smoothness prior, allows per-scene optimization to achieve impressive reconstruction results without leveraging any learned data priors (Zhang et al., 2020). However, as the optimization process can be poorly regularized and time-consuming, several methods have proposed using a monocular depth prior to guide the optimization process (Deng et al., 2022; LIU et al., 2025; Roessle et al., 2022). Furthermore, the introduction of 3D Gaussian splats (Kerbl et al., 2023) in recent years has led to methods that reconstruct either scene geometry (Liang et al., 2025b; Wu et al., 2024a; Yang et al., 2023b; LIU et al., 2025; Das et al., 2023), or both geometry *and* motion (Li et al., 2024a; Lin et al., 2024; Luiten et al., 2024; Yang et al., 2024b; Kratimenos et al., 2024; Duan et al., 2024), explicitly. Thus, a recent class of methods has additionally used 2D motion trajectories from pre-trained models to initialize and supervise the optimization of a scene-specific global representation (Liang et al., 2025a; Lei et al., 2024; Wang et al., 2024b). However, these methods use learned priors primarily to regularize the optimization and to reason about the visible parts of the scene. As such, while they generate more stable reconstructions in these visible regions, they perform no better than previous methods on view extrapolation tasks.

**Generative 4D reconstruction.** Given the extremely high number of images required to meet the Nyquist sampling rate for view interpolation from multi-view images (Chai et al., 2000; Mildenhall et al., 2019), and the fundamentally under-constrained nature of the view extrapolation problem, many works have explored the use of learned data priors to circumvent these theoretical limitations (Chen et al., 2024; Yinghao et al., 2024; Zhang et al., 2024b; Zhou et al., 2018). In the extreme case, these methods aim for view synthesis from a single RGB image. The most effective prior for this task, in terms of output quality, turns out to be a learned conditional probability distribution over the space of natural images. This is represented as a deep neural network that generates samples from the underlying distribution (Cao et al., 2024; Goodfellow et al., 2014; Xiong et al., 2024). Among this class of "generative" methods, Denoising Diffusion Probabilistic Models (DDPM) (Ho et al., 2020) have become especially popular due to their stability, mode coverage, quality, and flexibility (Croitoru et al., 2023). Consequently, a number of recent works have used diffusion to set the bar of quality on 3D reconstruction and view synthesis from single or sparse images (Yu et al., 2024; Gao* et al., 2024; Liu et al., 2024; Wu et al., 2023; Wang et al., 2024a). Subsequent work has extended the basic approach to 4D reconstruction from single images (Sun et al., 2024; Xul et al., 2024; Zhao et al., 2025), sparse image sets (Chou et al., 2025; Zhao et al., 2025), and from monocular videos (Ren et al., 2025; Van Hoorick et al., 2024; Zhang et al., 2024a). These methods rely on sampling all output frames from the generative model. Consequently, they do not provide accurate control over the camera movement and lack explicit spatio-temporal consistency constraints. In addition, the high memory requirements of video diffusion models limit these methods to short video sequences.

**Concurrent work.** CAT4D (Wu et al., 2024b) and Vivid4D (Huang et al., 2025) are concurrent works that show the use of generative models to improve per-scene optimized monocular reconstructions. Both works proceed by first generating multi-view video from a monocular sequence, which is used to supervise the optimization of a dynamic scene representation based on 3DGS. As such, the generation and optimization process is strongly decoupled. Our method uses an iterative approach, in which the generation steps alternate with the training of a Gaussian-based global 4D representation.

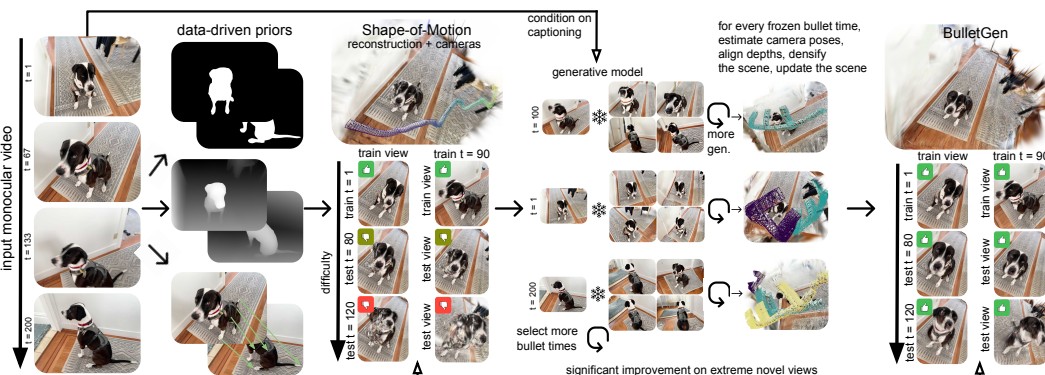

Figure 2: **BulletGen architecture.** Starting from a monocular RGB video, we reconstruct the dynamic scene with Shape-of-Motion, given data-driven priors (motion masks, depths, long-term 2D tracks). Then, we generate novel views at selected frozen time stamps (bullet times) using a conditioned generative model. These generated views are localized and mapped to the current scene using an optimization based on photometric, perceptual, semantic, and depth errors. The final 4D reconstruction augments the scene and allows for higher quality extreme novel view synthesis.

## 3 METHOD

Given $N$ frames of a monocular video sequence $\{I_t\}_{t=1}^{N}$ as input, our method acquires an initial 4D reconstruction of the scene as dynamic Gaussian splats. It then uses diffusion-based generative augmentation to obtain new observations for under-constrained regions, as well as synthesizing unseen parts of the scene. Using the original frames along with the generated views as supervision, the initial reconstruction is robustly optimized to obtain a photorealistic and globally consistent dynamic scene representation that allows for rendering of novel views at arbitrary time stamps and camera positions. Fig. 2 provides an overview of our pipeline.

### 3.1 INITIAL DYNAMIC GAUSSIAN SPLATTING RECONSTRUCTION

To bootstrap the initial reconstruction, we build on top of several state-of-the-art methods. We choose dynamic Gaussian Splatting as a scene representation, which is an extension to the original 3D Gaussian Splatting (3DGS) method. In particular, we use Shape-of-Motion, which relies on a range of scene priors extracted using methods like Track-Anything (Kirillov et al., 2023; Yang et al., 2023a) to extract masks for the moving objects, Depth Anything (Yang et al., 2024a) to estimate relative monocular depth maps, UniDepth (Piccinelli et al., 2024; 2025) for metric depth alignment, and TAPIR (Doersch et al., 2023; 2024) for obtaining long-range 2D tracks of moving objects.

Each 3D Gaussian is represented by RGB color $c \in [0,1]^3$, its center position $\mu(t) \in \mathbb{R}^3$, rotation $r(t) \in \mathbb{R}^4$ (quaternion), scale $s \in \mathbb{R}^3$ and opacity $o \in [0,1]$. For static Gaussians, $\mu(t)$ and $r(t)$ are constant for all time stamps. For dynamic Gaussians, they are defined as a time-dependent weighted combination of a global set of motion bases that are learned during optimization (usually a low number, *e.g.*, 20). Standard 3DGS (Kerbl et al., 2023) can be used to render the dynamic Gaussians for a given time stamp $t$. Given the required camera pose, all Gaussians are sorted by distance from the camera center and then rendered by alpha-composing the splatted 2D projections, which are affine approximations of the exact 3D projection. Given the complexity and ill-posedness of the task due to limited observations, we do not model view-dependent illumination changes with spherical harmonics. In the initial bootstrapping phase, we run Shape-of-Motion for 1000 optimization epochs to obtain a dynamic scene reconstruction without any generative component (Fig. 2).

### 3.2 GENERATIVE AUGMENTATION

After initial scene reconstruction, we generate a set $\{G_k^t\}_{k=1}^{K}$ of novel views of the scene at selected times $t$, and select a conditioning view and target motion (explained below). We generate novel views of the dynamic scene at bullet time $t$, assuming that the scene is frozen in time. We use an internal controllable image-to-video diffusion model that is trained to create novel views of the static

scene given a single image and a text prompt, which is generated by Llama3 (Grattafiori et al., 2024). This model follows state-of-the-art models in the field and is trained on standard static datasets. The conditioning prompt is a highly descriptive image captioning from the input video, which serves as an anchor to be consistent with the input. The used generative model also controls the motion by training different models for target motions, such as left and up directions. Then, the generated images $\{G_k^t\}_{k=1}^K$ are localized in the scene and used to update the scene with new observations.

**Camera tracking.** We estimate initial relative camera poses between frames of the same bullet time using the recently introduced VGGT model (Wang et al., 2025). Since VGGT-predicted depth is inaccurate, we estimate state-of-the-art monocular depth from MoGe (Wang et al., 2024c) and align it to VGGT depth. As MoGe/VGGT depths are not metric, we estimate a single scaling factor to align them to the current 4D reconstruction by minimizing the depth and RGB reprojection error in the covisible regions, producing final depth estimates $\{D_k\}_{k=1}^K$. At this stage, the camera poses of generated views are roughly aligned to the current scene reconstruction, but the alignment is not yet pixel-perfect, which is crucial for the following scene augmentation. To achieve pixel-perfect alignment, we leverage the Gaussian Splatting SLAM method SplaTAM (Keetha et al., 2024) for accurate camera tracking. Camera intrinsics for the generated cameras are assumed to be equal to the estimated intrinsics of the original monocular video. Extrinsics $\mathbf{E}_k \in \mathbb{R}^{4 \times 4}$ for generated views $G_k^t$ are initialized to the scale-aligned VGGT estimates. Then, the scene is rendered using the rendering function $\mathcal{R}(\mathbf{E}_k)$ producing RGB images $\mathcal{R}_I(\mathbf{E}_k)$ and depth images $\mathcal{R}_D(\mathbf{E}_k)$. Additionally, we render silhouettes $\mathcal{R}_S(\mathbf{E}_k)$ by accumulating Gaussian densities along the ray, which are used to define visibility masks as $V_k = \mathcal{R}_S(\mathbf{E}_k) > 0.99$.

**Robust loss and optimization.** The minimized loss function consists of four terms representing the photometric, perceptual, semantic, and depth errors. The photometric loss is the L1 loss in the RGB space, the perceptual loss is the LPIPS loss (Zhang et al., 2018) based on AlexNet (Krizhevsky et al., 2012) features, and the semantic loss is the cosine similarity between CLIP scores (Radford et al., 2021) of the renderings $\mathcal{R}_I(\mathbf{E}_k)$ and the generated images $G_k$. Finally, the depth loss is the L1 distance between depth renderings and the aligned depth maps. Since the generated images are usually not perfectly 3D consistent in the pixel space, we put the highest weight on the semantic and perceptual losses and keep the photometric loss only as a low-level learning signal. The final loss function is the weighted sum:

$$
\mathcal{L}(\{\mathbf{E}_k\}|V_k) = \sum_{k=1}^{K} \alpha_1 \text{L1}\Big(G_k, \mathcal{R}_I(\mathbf{E}_k)\Big|V_k\Big) + \alpha_2 \text{LPIPS}\Big(G_k, \mathcal{R}_I(\mathbf{E}_k)\Big|V_k\Big)
$$
$$
+ \alpha_3 \text{CLIP}\Big(G_k, \mathcal{R}_I(\mathbf{E}_k)\Big|V_k\Big) + \alpha_4 \text{L1}\Big(D_k, \mathcal{R}_D(\mathbf{E}_k)\Big|V_k\Big),
\tag{1}
$$

where we compute the loss only over the visible area $V_k$. We optimize camera poses $\{\mathbf{E}_k\}$ by minimizing the loss for 100 epochs, starting from the initialization described above. All other parameters, including the scene representation, are fixed. Once the camera poses are optimized, producing $\{\mathbf{E}_k^*\}$, it mostly leads to sufficiently accurate, pixel-wise alignment. To remove bad estimates, we keep only the first $K'$ generated views up until the first generative view whose loss $\mathcal{L}$ equation 1 is above a threshold $\gamma$, which is set in a conservative way so only well-aligned views are retained. After this step, we have $K' \leq K$ generated views $\{G_k\}$, their depths $\{D_k\}$, and optimized camera poses $\{\mathbf{E}_k^*\}$.

**Densification.** After obtaining well-aligned generative views, we now proceed to updating the scene. First, we perform Gaussian densification based on the densification mask proposed in SplaTAM (Keetha et al., 2024):

$$
M_k = \Big(\mathcal{R}_S(E_k^*) < 0.5\Big) + \Big(D_k < \mathcal{R}_D(E_k^*)\Big)\Big(L_1(D_k, \mathcal{R}_D(E_k^*)) < \lambda \text{MDE}\Big),
\tag{2}
$$

where we densify areas with insufficient density or where new geometry is in front of the current geometry, unless it is due to noise as measured by $\lambda = 50$ times the median depth error (Keetha et al., 2024). This way, we initialize a new Gaussian at every pixel in the densification mask based on its color and depth. For the newly densified Gaussians, we decide if they are static or dynamic based on the nearest neighboring Gaussian's static/dynamic label, where the dynamic Gaussians' locations are moved to the currently generated bullet time stamp. The weights for the motion bases of new dynamic Gaussians are initialized to the nearest neighbor.

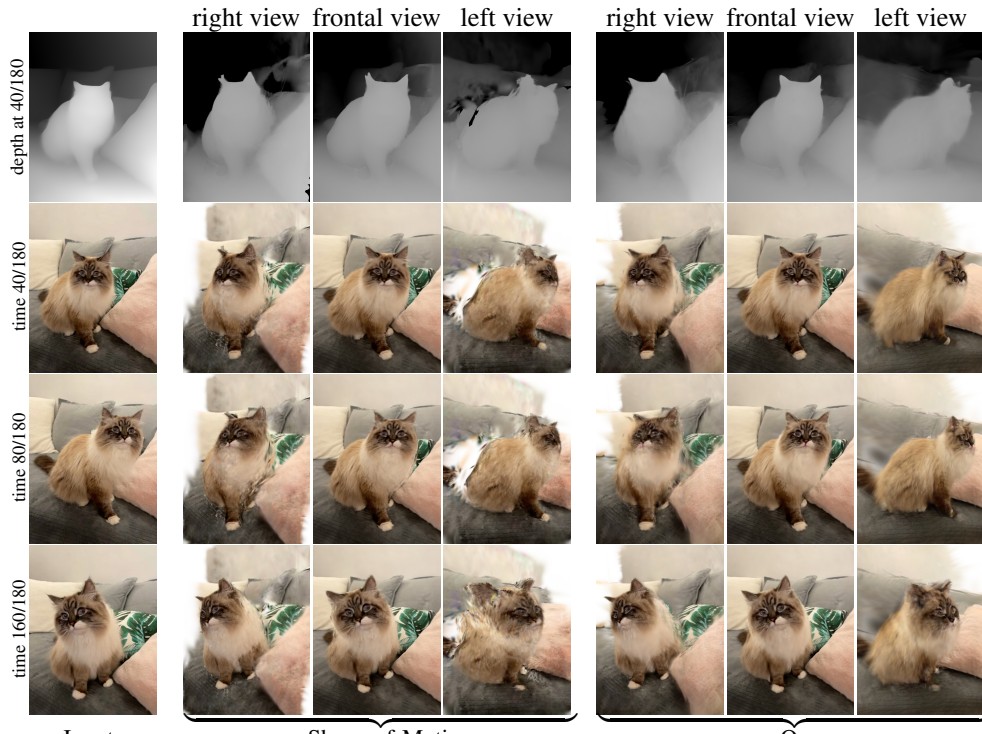

Figure 3: **Extreme novel view synthesis across space and time.** The generation for training was performed $n_G = 7$ times for $n_S = 5$ bullet-time stamps, *i.e.* 1, 45, 90, 135, 180 for a sequence with 180 frames. The renderings shown here are at time stamps and viewpoints that were not generated using a generative model, which shows that using only several bullet-time reconstructions is enough to reconstruct a dynamic scene. Temporal slices for all 180 time stamps are shown in Fig. 5.

**Scene update.** To update the reconstructed scene, we optimize a weighted joint loss, which is a sum of the proposed tracking loss $\mathcal{L}$ equation 1 on the generated views, including all previous generated views over all bullet time stamps, and the SoM loss function to always keep the scene consistent with the original video. Since the scene is already densified, we do not need the visibility masks $V_k$ in this stage and, thus, the loss is computed over the entire generated image, as compared to the loss used for camera tracking. We optimize this joint loss function for 100 epochs by optimizing parameters of all Gaussians and the motion bases. After this, we repeat it all with new generated views (Fig. 2).

**Time selection.** One of the hyperparameters of our method is the number $n_S$ of sampled bullet-time stamps. We sample them uniformly between the first and last time stamps, *i.e.* $t = 1$ and $t = N$. For robust estimation, we start with the middle frame bullet-time stamp, *e.g.* $N/2$. Then, we proceed with the furthest time stamps. For example, when $N = 100$ and $n_S = 5$, the sampled time stamps are $\{1, 25, 50, 75, 100\}$, but in the order $\{50, 1, 100, 25, 75\}$. Also note that the bullet-time stamp selection can be progressive and proceed auto-regressively, and in our example, can continue with additional time stamps $\{12, 37, 62, 87\}$ for $n_S = 9$.

**View selection.** The used generative model currently supports only static scenes and comes with two operating modes for generating left- and upward trajectories. By flipping the image horizontally, applying the leftward model, and flipping back, we can also generate rightward trajectories. Note that we do not apply the same principle to the upward direction, because the model only works well with upright imagery. Thus, we select between three modes for the view selection, *i.e.* left, right, up. Once bullet time is selected, we proceed with $n_G$ generations for the selected time. We tested three modes. First, for $n_G = 3$, we select $\{$up, left, right$\}$ in this order. For $n_G = 5$, we select $\{$up, left, right, left, right$\}$, and for $n_G = 7$, the used sequence is $\{$up, left, up, right, up, left, right$\}$. The view selection for generative model conditioning for each chosen direction is as follows. For leftward motion, we choose the left-most image (either the original one or the generated one), where the angle is measured from the middle of the original camera poses and viewing at the middle of the dynamic scene. Similarly, for rightward motion, we choose the right-most image. In all cases, only images at the currently selected bullet-time stamp are considered, either generated or the original one (there

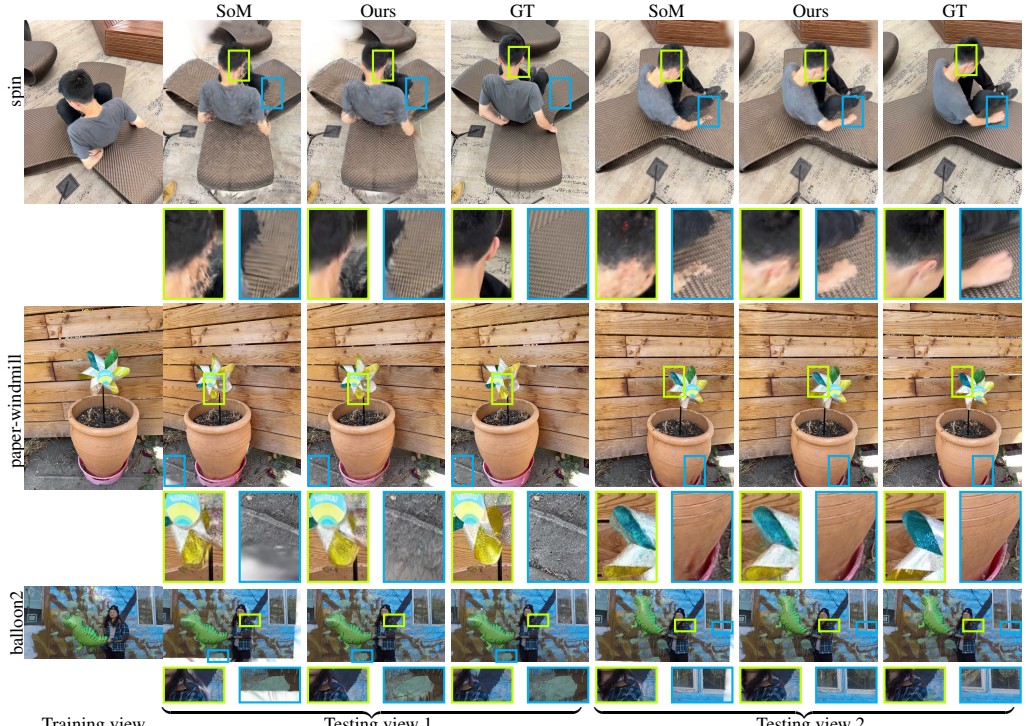

Figure 4: **Qualitative evaluation on several benchmark datasets.** The input is a monocular video as shown in the training view column. Both benchmark datasets (Nvidia and DyCheck iPhone) have several additional testing cameras. We compare novel view synthesis to Shape-of-Motion and zoom-in to highlight the differences. Our method is able to provide more accurate and sharper reconstructions, both in static and dynamic parts of the scene.

is only one original image for every time stamp). For upward motion, we choose the last generated image at this bullet time (or the original one if this is the first generation).

**Implementation details.** All optimizations are performed using the ADAM optimizer (Kingma & Ba, 2017). The batch size for the generative views and the original video frames is set to 8 (thus, we have two concatenated batches of 8 in every iteration). The hyperparameters in equation 1 have been empirically determined and fixed to $\alpha_1 = 0.02$, $\alpha_2 = 0.1$, $\alpha_3 = 0.1$, $\alpha_4 = 0.5$ for all experiments. The threshold for generative views is set to $\gamma = 0.4$. For the SoM loss function on the original video, we use their default weights. The total number of generated views per generation is fixed at $K = 50$. The number of generations used is $n_G = 7$, while the number of selected bullet times is $n_S = 9$, unless specified otherwise. The average optimization time on a sequence from the iPhone dataset with full resolution takes around 3 hours (including initial SoM optimization of 1.5 hours) on an Nvidia A100 80GB GPU. All main experiments required around 1 week of A100 GPU time, and preliminary non-included experiments required an additional month of GPU time. For evaluation on custom data, we use MegaSaM (Li et al., 2024b) to estimate camera poses. For evaluation on benchmark datasets, we used provided poses computed from COLMAP (Schönberger & Frahm, 2016; Schönberger et al., 2016) for fair comparison to other methods.

## 4 EXPERIMENTS

We evaluate on the DyCheck iPhone (Gao et al., 2022) and Nvidia dynamic (Yoon et al., 2020) datasets, which are commonly used datasets for novel view synthesis evaluation in dynamic scenes.

**View synthesis metrics.** We evaluate the standard PSNR, SSIM, and LPIPS scores between the ground truth and the rendered novel views. On the iPhone dataset, we measure those metrics on the covisible regions for fair comparison to other methods. As mentioned by Liang et al. (2025a), these scores are not robust enough on the current datasets due to camera misalignment, color differences

| Method | 3D Tracking | | | 2D Tracking | | |
|---|---|---|---|---|---|---|
| | EPE$\downarrow$ | $\delta_{3D}^{.05}\uparrow$ | $\delta_{3D}^{.10}\uparrow$ | AJ$\uparrow$ | $<\delta_{avg}\uparrow$ | OA$\uparrow$ |
| HyperNeRF | 0.182 | 28.4 | 45.8 | 10.1 | 19.3 | 52.0 |
| DynIBaR | 0.252 | 11.4 | 24.6 | 5.4 | 8.7 | 37.7 |
| Deformable-3D-GS | 0.151 | 33.4 | 55.3 | 14.0 | 20.9 | 63.9 |
| CoTracker + DepthAnything | 0.202 | 34.3 | 57.9 | 24.1 | 33.9 | 73.0 |
| TAPIR + DepthAnything | 0.114 | 38.1 | 63.2 | 27.8 | 41.5 | 67.4 |
| Shape-of-Motion | 0.082 | 43.0 | 73.3 | 34.4 | 47.0 | 86.6 |
| Ours | **0.071** | **51.6** | **77.6** | **36.6** | **49.5** | **87.4** |

Table 1: **Evaluation on the iPhone dataset,** 3D and 2D tracking. Our method outperforms all state-of-the-art methods in terms of 2D and 3D tracking accuracy as measured by various metrics.

| Method | PSNR$\uparrow$ | SSIM$\uparrow$ | LPIPS$\downarrow$ | CLIP-I$\uparrow$ |
|---|---|---|---|---|
| T-NeRF | 15.60 | 0.55 | 0.55 | 0.86 |
| HyperNeRF | 15.99 | 0.59 | 0.51 | 0.87 |
| Deform.-3DGS | 11.92 | 0.49 | 0.66 | 0.79 |
| SoM | 16.72 | 0.63 | 0.45 | 0.86 |
| HiMoR | - | - | 0.46 | 0.89 |
| CAT4D (no code) | **17.39** | 0.61 | **0.34** | - |
| Ours | 16.78 | **0.64** | 0.39 | **0.90** |

| Method | PSNR$\uparrow$ | SSIM$\uparrow$ | LPIPS$\downarrow$ |
|---|---|---|---|
| 4D-GS | 14.01 | 0.39 | 0.59 |
| CoCoCo | 14.99 | 0.47 | 0.53 |
| StereoCrafter | 14.85 | 0.49 | 0.57 |
| ViewCrafter | 14.94 | 0.49 | 0.58 |
| SoM | 14.56 | 0.46 | 0.53 |
| Vivid4D (no code) | 15.20 | 0.50 | 0.49 |
| Ours | **16.38** | **0.51** | **0.45** |

Table 2: **Evaluation on the iPhone dataset**, novel view synthesis. The proposed method achieves state-of-the-art performance in terms of novel view synthesis.

Table 3: **Evaluation on the iPhone dataset**, subset chosen by Vivid4d. Our method outperforms all other methods.

between the training and test cameras, as well as the nature of the task. Thus, we also report CLIP-I (Radford et al., 2021) scores, which we see as more representative in this ill-posed task.

**Tracking metrics.** The iPhone dataset contains long-range 3D point tracking annotations. Thus, we also measure 3D end-point-error (EPE) at every timestep and the percentage of points that are within a certain radius of the ground truth 3D point (denoted by $\delta_{3D}^{\tau}$, where $\tau$ is either 5cm or 10cm). By projecting those 3D tracking annotations, we obtain 2D tracking annotations and report Average Jaccard (AJ) metric, average position accuracy ($<\delta_{avg}$) and Occlusion Accuracy (OA) metric.

**Baselines.** We compare to a range of methods that are specifically designed for dynamic scene reconstruction. Methods based on Neural Radiance Fields (NeRF) (Mildenhall et al., 2020) and its variants are T-NeRF (Li et al., 2023a), HyperNeRF (Park et al., 2021b), and DynIBaR (Li et al., 2023b). More recently, methods based on Gaussian Splatting were proposed, such as Deformable-3DGS (Yang et al., 2023b), 4D-GS (Wu et al., 2024a), Shape-of-Motion (Wang et al., 2024b), and HiMoR (Liang et al., 2025a). We also compare to methods based on generative diffusion models, such as CoCoCo (Zi et al., 2024), StereoCrafter (Zhao et al., 2024), ViewCrafter (Yu et al., 2024), CAT4D (Wu et al., 2024b), and Vivid4D (Huang et al., 2025). Many of these methods are concurrent with our work and do not provide code, *e.g.* CAT4D and Vivid4D. Nevertheless, we strive to provide insightful comparisons by evaluating on the reported datasets chosen by these methods. For 2D and 3D tracking scores, we additionally compare to CoTracker (Karaev et al., 2024) and TAPIR (Doersch et al., 2023), which are lifted to 3D by DepthAnything (Yang et al., 2024a).

**The iPhone dataset.** The iPhone dataset contains 12 sequences with a moving training camera captured with hand-held iPhone device, two static test cameras (for 5 sequences), and 3D tracking annotations. As shown in Table 1, our method outperforms all other state-of-the-art methods on all metrics in terms of 2D and 3D tracking scores. This is achieved by providing more constraints for scene geometry, which allows for better 3D triangulation during scene reconstruction. We also achieve state-of-the-art scores in terms of novel view synthesis (Table 2). We also add results reported in CAT4D (no code), which achieves slightly better PSNR and LPIPS scores, whereas our method obtains better SSIM results. However, we want to stress that the table remains incomplete in terms of CLIP-I score due to a lack of publicly accessible code and missing information about the training

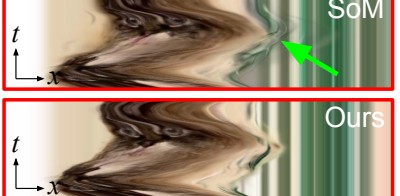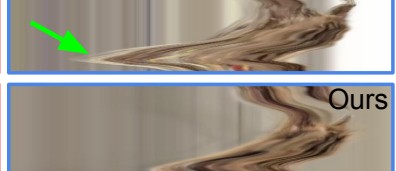

Figure 5: **Temporal plane slices of extreme camera views.** Visualizing the highlighted rows across time in $xt$ space shows that our method suffers from fewer temporal artifacts than Shape-of-Motion (SoM) caused by floating Gaussians and exploding geometry. Our rendered view is shown on the left.

procedure and its overall training scale. To compare to Vivid4D, we choose their challenging subset of the iPhone dataset. Our method again outperforms other methods (Table 3). This highlights that our method is more effective in those challenging cases (as compared to the full dataset).

**The Nvidia dataset.** The Nvidia dataset contains 9 dynamic scenes captured by 12 synchronized static cameras. We use camera 1 as input, and cameras 2 and 3 as output for evaluation. This dataset contains testing views close to the training views, therefore the CLIP-I and LPIPS metrics do not show significant improvement, but in terms of PSNR, our improvement w.r.t. Shape-of-Motion is significant (Table 4).

| | PSNR↑ | SSIM↑ | LPIPS↓ | CLIP-I↑ |
|---|---|---|---|---|
| SoM | 15.26 | 0.454 | 0.388 | **0.87** |
| Ours | **17.02** | **0.462** | **0.386** | 0.87 |

Table 4: **Evaluation on the Nvidia dataset.** Our method improves the Shape-of-Motion (SoM) reconstruction on all metrics.

**Ablation study.** The main hyperparameters of our method are the number of generations $n_G$ per selected bullet-time stamp, and the total number of those time stamps $n_S$. Table 5 shows that increasing $n_G$ per time stamp slightly improves most metrics, but is not significant given that the test cameras are not far from the training ones. A high value for $n_G$ is important for extreme novel view point synthesis. However, the number of selected bullet time stamps $n_S$ is vital even in the standard setting, and a higher number of time stamps improves all metrics.

| Method | PSNR↑ | SSIM↑ | LPIPS↓ | CLIP-I↑ |
|---|---|---|---|---|
| SoM | 16.72 | 0.63 | 0.45 | 0.86 |
| $n_s = 9, n_g = 3$ | **16.81** | 0.63 | 0.40 | 0.89 |
| $n_s = 9, n_g = 5$ | 16.80 | 0.63 | 0.40 | 0.89 |
| $n_s = 9, n_g = 7$ | 16.78 | **0.64** | **0.39** | **0.90** |
| $n_s = 3, n_g = 7$ | 15.95 | 0.62 | 0.45 | 0.88 |
| $n_s = 5, n_g = 7$ | 16.36 | 0.62 | 0.44 | 0.89 |
| $n_s = 9, n_g = 7$ | **16.78** | **0.64** | **0.39** | **0.90** |

Table 5: **Ablation study** (iPhone dataset). The number of generations $n_G$ per time stamp slightly improves most metrics, and the number of selected bullet-time stamps $n_S$ improves all metrics.

**Qualitative results.** Qualitative results for the iPhone dataset are provided in Fig. 1 (cat, dog) and Fig. 4 (spin, paper-windmill), and for the Nvidia dataset in Fig. 1 (skating) and Fig. 4 (balloon2). As can be seen in Fig. 4, the testing views in the benchmark datasets are close to the training views. However, our method allows for much more extreme novel view synthesis, going beyond what is possible by other methods. Unfortunately, there is no dataset with such extreme view point testing cameras, which can be seen as a limitation in this emerging field. Fig. 3 shows qualitative examples of extreme novel views. BulletGen significantly improves rendering quality in such cases. Temporal slices in the $x$ direction are shown in Fig. 5, which highlights that our method produces smooth and consistent rendering results over time, especially when compared to SoM.

**Limitations.** BulletGen assumes that the initial Shape-of-Motion optimization performs reasonably well, at least from the viewpoints of the original video, which in turn depends on how well all monocular priors were estimated.

## 5    CONCLUSION

We introduced a method for dynamic 3D reconstruction from a monocular video, effectively addressing extreme novel view synthesis challenges with drastic improvements over the state of the art. By integrating static bullet-time video generation with dynamic 3D Gaussian splatting our approach enhances both 2D and 3D tracking accuracy, and novel view synthesis. Our method seamlessly incorporates new scene elements, improving rendering quality. Our results on standard benchmark datasets confirm the efficacy of this approach for dynamic scene reconstruction in complex environments.

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
