# OpenReview forum: "BulletGen: Improving 4D Reconstruction with Bullet-Time Generation"
_ICLR.cc/2026/Conference — ICLR 2026 Conference Withdrawn Submission_

### Official Review · Reviewer_EcZr · 2025-10-27

**Soundness:** 3
**Presentation:** 3
**Contribution:** 2
**Rating:** 2
**Confidence:** 4

**Summary:**

The paper proposes BulletGen, a method for dynamic 3D scene reconstruction from monocular videos that leverages a generative diffusion model to enhance 4D reconstruction. The key innovation is using the static image-to-video diffusion models to generate novel views at a selected timestep, which are then aligned to the scene representation and used for supervision. The method benchmarks its performance on DyCheck and Nvidia datasets.

**Strengths:**

- The main strength of the method is the proposed iterative approach that allows for iterative refinement of the Gaussian scene representation.
- The use of a novel view generation diffusion model, given a static scene, is a nice way of using a static model for dynamic reconstruction.
- The proposed alignment algorithm seems to be working well. Authors make a really good use of the state-of-the-art models for priors.
- The provided evaluation shows consistent improvement over the baseline Shape-of-Motion.
- The qualitative results align with the quantitative ones in terms of visible improvements with respect to Shape-of-Motion.

**Weaknesses:**

- A big part of the paper's contribution is dependent on the 'internal controllable image-to-video diffusion model'. This raises several concerns. Firstly, the reproducibility of the method will be largely limited unless the model is released. Any further comparison with this method will highly likely not be feasible. While it is fair to use an internal model to achieve a good performance, in my opinion, it should be accompanied by a detailed comparison of the same pipeline but with a publicly available diffusion model. This is particularly important given the fact that authors use ViewCrafter in their experiments. There is nothing in the paper suggesting that using ViewCrafter would not be feasible in this setup. In the current state, it is not clear how much of the performance improvement comes from the diffusion model.
- The novelty of the paper is more limited than the authors mention; several papers with a rather similar approach are not mentioned, and the evaluation is missing important comparisons.
	- Regarding novelty - I believe two important references are missing. Firstly, Difix3D+ [1] is an important work that proposed an iterative refinement process of the 3D scene with the use of a generalisable enhancement diffusion model. Not only does this work propose a significant method in part similar to this paper's contributions, but the diffusion model is available and could be used in an ablation study. Further, ViDAR [2] proposes a reconstruction method in which the novel views are generated and further enhanced with personalised diffusion to serve as the reconstruction supervision. This work uses a similar idea of generating an additional supervision signal in novel views and seems highly relevant as a related work.
	- Regarding evaluation - the authors cite MoSca in line 133; the evaluation should include MoSca as the compared method. It looks like MoSca would outperform BulletGen in some metrics (PSNR, SSIM). Further, the aforementioned ViDAR could be included as well (performing better in PSNR, SSIM, LPIPS). Whilst the work recently got accepted to NeurIPS, and it seems not to have released the code yet, the arXiv release includes the numerical results on DyCheck in the same setup as in this paper, which makes it the same comparison as the Vivid4D used here.
- In terms of ablation, it would be good to see isolation of contributions, i.e., add contributions to Shape-of-Motion one by one to show the reader the importance of each. This would help with the previously raised point of not being able to isolate the impact of the internal model.
- To strengthen the claim on the new synthesised plausible parts of the scene, one could show that both quantitatively and qualitatively. Namely, given that DyCheck provides covisibility masks, presenting results outside of such masks would effectively measure the performance of the approach in unseen parts of the scene.
- CAT4D is currently way past the date of being a concurrent work; it was published at CVPR 2025 and released on arXiv even earlier.
- It would be good to see examples of the captions and whether they differ between times and views.

[1] Jay Zhangjie Wu, Yuxuan Zhang, Haithem Turki, Xuanchi Ren, Jun Gao, Mike Zheng Shou, Sanja Fidler, Zan Gojcic, Huan Ling, *Difix3D+: Improving 3D Reconstructions with Single-Step Diffusion Models*, CVPR 2025

[2] Michal Nazarczuk, Sibi Catley-Chandar, Thomas Tanay, Zhensong Zhang, Gregory Slabaugh, Eduardo Pérez-Pellitero, *ViDAR: Video Diffusion-Aware 4D Reconstruction From Monocular Inputs*, NeurIPS 2025

**Questions:**

- Could you generate all your novel views with the diffusion model from the input video? In this way, the prior provided to the diffusion model would be the strongest, as opposed to a degraded view produced by baseline reconstruction. It would be an interesting ablation.
- Regarding one of the weaknesses, can any diffusion be used in the pipeline?
- In higher numbers of n_g, what is the rationale of doing multiple generations the same way?
- It sounds like, given a timestep, an extreme view among input poses is selected as the prior for generation. Therefore, for some timesteps, the view will be close to the input, and for some, very far. For the far views, the prior (i.e. novel view) will highly likely contain some artefacts. This would, in turn, introduce a likely noisy supervision to the reconstruction model. Did you observe anything like that? Is there a way to mitigate that?
- Regarding time complexity, could you report a time for generating one sequence of novel views given the input image for the diffusion model?

---

### Official Review · Reviewer_27FN · 2025-11-01

**Soundness:** 3
**Presentation:** 3
**Contribution:** 2
**Rating:** 4
**Confidence:** 3

**Summary:**

- BulletGen addresses the ill-posed problem of 4D reconstruction from monocular videos by leveraging a diffusion-based video generation model to correct errors and complete missing information in Gaussian-based dynamic scene representations.
- The method aligns generated frames at specific "bullet-time" stamps with the initial 4D reconstruction to supervise and "iteratively" optimize the dynamic 4D Gaussian model using a robust loss incorporating photometric, perceptual, semantic, and depth err.

**Strengths:**

- Generative augmentation for unobserved regions: The method employs a frozen diffusion-based image/video generator at "bullet-time" instants to hallucinate novel views (e.g., back sides, occluded areas), providing missing appearance and geometry cues.
- Integration of 2D generative priors with global 4D scene representation: Generated 2D frames undergo pose-tracking and depth-alignment, then iteratively supervise a dynamic 3D Gaussian-splatting representation to maintain spatio-temporal consistency.

**Weaknesses:**

- Since the method ultimately relies on diffusion loss, performance improvements are inevitably limited when dealing with complex objects or large motions. This can be confirmed by the modest gains observed in SoM. (Particularly, while there is some gain shown in Table 3, the 0.06dB improvement in Table 2 is too minimal to be considered significant.)
- Furthermore, the use of CLIP loss tends to work better primarily on object-centric scenes. Consequently, as observed, the method performs better on object-centric and near-rigid scenes such as "spin" and "paper-windmill," which represents a limitation.

**Questions:**

- I'm curious about how the performance would change if this approach were applied to MoSca, a more recent method.
- Since the generation ultimately depends on the prompt, which frame was used to select the prompt? For example, I understand that the prompt can vary quite significantly across frames when objects are moving.
- Was the generative model not fine-tuned?

---

### Official Review · Reviewer_pJEB · 2025-11-01

**Soundness:** 4
**Presentation:** 3
**Contribution:** 2
**Rating:** 6
**Confidence:** 4

**Summary:**

This paper presents BulletGen, a method for 4D dynamic scene reconstruction from a single monocular video. The core problem it addresses is the highly ill-posed nature of this task, particularly in reconstructing unseen regions and resolving depth ambiguities.
The authors demonstrate that this approach achieves state-of-the-art results on the DyCheck iPhone and Nvidia datasets, significantly improving novel-view synthesis quality and 2D/3D tracking accuracy.

**Strengths:**

1. Strong Empirical Results: The method shows impressive quantitative and qualitative results. It achieves state-of-the-art performance on standard benchmarks for both novel-view synthesis and 2D/3D tracking, clearly outperforming its baseline (Shape-of-Motion) and other recent methods. The qualitative results for extreme novel views (e.g., Fig. 1 and Fig. 3) are particularly strong.
	2. Simple but effective strategy. A significant strength is the method's practicality. Instead of requiring a complex, computationally expensive 4D video diffusion model, the authors leverage a generator trained only on static scenes. This "bullet-time static diffusion strategy" is a clever way to augment a dynamic reconstruction while using more accessible and abundant static training data

**Weaknesses:**

1. Limited Novelty of the Core Concept: The central idea of using diffusion models to generate novel views as supervision for a 3D/4D neural representation is not new. This concept has been substantially explored in the 3D reconstruction domain, particularly for sparse-view inputs (e.g., ReconFusion [1], Difix+[2], and others ). The paper's primary contribution is the application of this idea to the monocular 4D setting. While effective, this can be seen as an incremental, though logical, extension of existing work
	2. Unconvincing "Bullet-Time" Strategy and Temporal Consistency: My main concern lies with the "bullet-time" generation. The generative model is static and, more importantly, generates novel views for each time stamp independently. The paper's claim is that optimizing the global 4DGS representation (which uses shared motion bases ) is sufficient to enforce temporal consistency across these independently generated views.
	Other works in 4D object generation (e.g., EG4D [3], SV4D [4]) have proposed more principled solutions, such as attention-based mixing or latent-space temporal models, to explicitly enforce consistency during the generation phase. The paper lacks a discussion or comparison against such temporally-aware generative methods. The provided temporal slice (Fig. 5) only compares against the SoM baseline, which is insufficient to prove the temporal coherence of the generated content.

	[1] Wu et al. CVPR 2024.
	[2] Zhang et al. CVPR 2025.
	[3] Sun et al. ICLR 2025.
	[4] Xie et al. ICLR 2025.

**Questions:**

1. The justification for using Shape-of-Motion (SoM) as the baseline is unclear. Given that SoM itself has notable limitations, could the authors elaborate on why this particular model was chosen over other, stronger 4D reconstruction methods?

---

### Official Review · Reviewer_q93b · 2025-11-02

**Soundness:** 2
**Presentation:** 2
**Contribution:** 2
**Rating:** 4
**Confidence:** 4

**Summary:**

This paper presents BulletGen, a method for 4D dynamic scene reconstruction from monocular videos. Its core contribution is the integration of static, diffusion-based bullet-time generation with dynamic 3D Gaussian Splatting to address under-constrained regions. The approach iteratively augments the scene representation at selected frozen timestamps. Evaluations on the DyCheck iPhone and Nvidia Dynamic datasets demonstrate state-of-the-art performance in novel view synthesis and 2D/3D tracking.

**Strengths:**

1. the generation steps alternate with the training of a Gaussian-based global 4D representation.
2. The paper provides thorough quantitative evaluations on multiple datasets, demonstrating improvements in metrics like PSNR, SSIM, LPIPS, and tracking accuracy. The ablation study effectively analyzes the contributions of key hyperparameters, such as the number of bullet-time stamps and generations.

**Weaknesses:**

1. A similar idea has already been employed in 3D reconstruction works like VistaDream, which also involves initializing a 3D Gaussian Splatting (3DGS) reconstruction and then iteratively inpainting it. This paper should further discuss its relationship with such literature.
2. The outputs of diffusion models often exhibit insufficient 3D coherence, which can complicate the alignment process and introduce visual artifacts, especially when handling complex dynamic scenarios.
3. The description of key technical components lacks clarity. The explanation of the generative augmentation pipeline and the loss function, while detailed, suffers from unnecessary complexity. It often fails to provide a clear intuition for design choices. For example, the iterative optimization loop and the mechanism for integrating generated views into the global 4D representation are not explained in a coherent, step-by-step manner, making the core contribution difficult to follow.

**Questions:**

My main consider are the novelty of involving the video gen model to 4d reconstruction and detailed technique contribuitions.

---

### Note · Authors · 2025-11-14

I have read and agree with the venue's withdrawal policy on behalf of myself and my co-authors.